# FrogVLE Application in Science Teaching in Secondary Schools in North Malaysia: Teachers' Perspective

**Nofaizah Ramli** 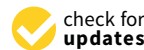 **and Salmiza Saleh** *

School of Educational Studies, Universiti Sains Malaysia, Penang 11800, Malaysia; faizahramli2438@gmail.com
* Correspondence: salmiza@usm.my

**Abstract:** The frogVLE application was launched in Malaysia to provide a virtual learning environment in order to produce competitive and relevant students in the 21st century. As science is one of the most important subjects for the development of a nation, the perspective of teachers as policy implementers should be taken into account in ensuring that the desire is achieved. Hence, this study aimed to investigate the level of application of frogVLE in secondary school science teaching in the northern peninsular of Malaysia. This qualitative study involved 50 secondary school science teachers in Perak, Kedah, Penang and Perlis. Each respondent was required to answer a questionnaire which consists of 20 questions. Five teachers were randomly interviewed by the researcher. The data were analyzed descriptively. The analysis results provide a real picture of how the frogVLE application is used in teaching, as well as the issues, challenges and suggestions for improvements in frogVLE while teaching secondary school science subjects. The findings from this study are expected to help the Ministry of Education to develop programs and improve the use of the frogVLE application in teaching science in secondary schools. In conclusion, a virtual learning environment such as the frogVLE application can only be used optimally to help improve science teaching excellence if the relevant issues are solved and supportis received from all parties.

**Keywords:** frogVLE; virtual learning environment; science teaching; secondary school; North Malaysia

## 1. Introduction

As the world has fundamentally changed in becoming the age of Knowledge Economy in the 21st century, the roles of learning and teaching must also change if education is to meet its moral purpose [1]. Therefore, the frogVLE application was launched in Malaysia to provide a virtual learning environment in order to produce competitive and relevant students in the 21st century [2]. As science is one of the important subjects for the development of a nation, the perspective of Malaysian teachers as policy implementers should be taken into account in ensuring that the desire is achieved. FrogVLE [3] is actually a web-based learning system that replicates real-world learning by integrating virtual equivalents of conventional concepts of education. For example, teachers can assign lessons, tests, and marks virtually, while students can submit homework and view their marks through the virtual learning environment (VLE). Parents can view school news and important documents while school administrators can organise their school calendars and disseminate school notices via the Internet. A virtual learning environment (VLE) such as frogVLE is one of the mediums in mixed learning that emphasizes learning and teaching which consist of a mix of online learning approaches and face-to-face learning modes. About 30%–80% of content and learning activities are conducted online which supports or replaces face-to-face learning [4]. It allows learning and teaching to occur anywhere in accordance

with the learning of the 21st century, which is characteristically flexible and student-centered. A virtual learning environment helps teachers in managing student learning processes virtually [5]. The use of frogVLE creates an interactive learning environment that is capable of attracting students to science subjects. There are various applications that can be used, such as communication tools, tools for organizing the administration of the teaching process, and learning and student assessment tools [6,7]. In order to compete in this sophisticated 21st century world, the Government of Malaysia launched the 1bestarinet program in 2012 to provide high-speed Internet access and a virtual learning platform (frogVLE) to bridge the digital divide between urban and rural areas [2].

Previous studies indicated that frogVLE has a great effectiveness for both teachers and students [8,9]. FrogVLE has created great interest among students and easy access to a wealth of materials and resources. These teaching and learning resources can help to improve learning outcomes and increase self-directed learning among the students. Flexibility in learning regardless of place and time increases student motivation and in the process, also increases their information and communications technology (**ICT**) usage and awareness of ICT's potential as an alternative way of learning. This 'updated' approach of learning, in the long run provides greater opportunities when these students enter the job market [9]. As for the teachers, frogVLE helps make their teaching job easier than the traditional approach. In addition to being easy to use, the system also helps them to organize their teaching and learning materials. This saves them time when updating or locating specific materials. They are also pleased that with the system, they need not print or distribute handouts and this reduces cost substantially [9].

Support from multiple parties is necessary to ensure that frogVLE can be fully utilized. Teachers' ICT skills and ICT facilities are the major factors influencing the use of frogVLE among teachers in secondary schools [10,11]. However, studies have shown that most teachers have a modest level of knowledge and skill in that they are familiar with application software such as word processing and electronic presentations but not internet and email applications [11–13] found that teachers actively use their computers and smart phones in accessing a variety of online information and resources that help them in building their lesson plan and developing their lesson notes for students. However, the use of computer and internet in classroom teaching ison the decline [12]. Teachers often express their lack of confidence in their digital technology skills and this can implicitly affect their attitudes towards the use of digital technology in their teaching [14]. According to Mahizer and Mohd Azli [15], teachers basically receive the implementation of the learning process using the frogVLE platform, but technical constraints cause teachers to struggle when practicing it as the primary medium of student learning. Due to the constraints expressed as well as thehigh maintenance incurred by e-learning, some teachers insist on their preference for the traditional face-to-face approach.

Teachers' perceptions not only influence the thinking processes and decisions teachers make during their planning and interaction, they also have a significant impact on teaching behaviour. This implies that it is important for teachers to evaluate their own teaching beliefs, apply their self-identified strengths through actual teaching behaviour, and engage students in learning such that they perceive learning as enjoyable [16,17]. Teachers require a mindset that is best adapted to the significant changes in learning; teachers who develop strategies for engaging with and constructing new knowledge teach the future generation with the knowledge that it is unknown yet filled with possibilities. This means that teachers need to have growth rather than fixed mindsets [18]. Many studies have been made regarding the use of ICT in the classroom but the focus is not on frogVLE [8,12,19,20]. One study did look at the ICT level and teacher creativity but it only involved certain areas and specific items. Studies are conducted only in certain areas, whereas every school in Malaysia in general and in the north of the Peninsular Malaysia has a specific issue in the implementation of frogVLE. The different geographical factors of each state in Northern Malaysia (Perak, Kedah, Penang and Perlis) with the race and cultural diversity of teachers and students slightly influence the teaching in the classroom. Thus, each teacher must have a high level of cultural competency to ensure that the teaching and learning process can be conducted smoothly without being affected by the different backgrounds that exist

among students from various ethnic and cultural groups [21]. The data obtained are still inadequate to create proficiency level profiles and ICT proficiency, especially in the frogVLE application among secondary school science teachers in Northern Malaysia as references to all parties [12,20,22–24].

Teachers as policy implementers in the Ministry of Education play an important role in ensuring that frogVLE is optimally used in the teaching of science. Therefore, the purpose of this study was to get a view of science teachers in North Malaysia to know the level of frogVLE used in teaching science. Specifically, the research objectives were to provide a real picture of how the frogVLE application is used in teaching, as well as the issues, challenges and suggestions for improvements in frogVLE while teaching secondary school science subjects.In particular, the following research questions were addressed:

i. What is the frequency of using frogVLE among science teachers in North Malaysia while teaching secondary school science subjects?

ii. How much time is allocated each time to using frogVLE among science teachers in North Malaysia while teaching secondary school science subjects?

iii. What are the frequently used applications in frogVLE among science teachers in North Malaysia while teaching secondary school science subjects?

iv. What are the issues and challenges of frogVLE among science teachers in North Malaysia while teaching secondary school science subjects?

v. What are the suggestions for improvement in frogVLE among science teachers in North Malaysia while teaching secondary school science subjects?

## 2. Methodology

This qualitative study involved 50 secondary school science teachers randomly selected from the states of Perak, Kedah, Penang and Perlis. The teachers selected in this study were teachers who taught a secondary school science subject from Form 1 to Form 5. Each respondent was required to answer the questionnaire provided by the researcher. The questionnaire in the form of a Google Form wassent to all teachers in the north via email, social media and face-to-face. The length of time to answer the questionswas between 5 and 10 min. Prior to responding to the questionnaire, they were informed of the aim and importance of the study. When answering the survey, if there was an ambiguity in terms of understanding the questions, additional explanations were given to the respondents. The questionnaire, which is a Likert-type scale containing 20 items, was adapted from Mei et. al. [9]. The questionnaire consists of two parts: part A—demographic information and part B—frogVLE application. Part B is divided into four themes (time allocation, application, issue and challenge, and suggestions) which contain five items each. Content validity was done in order to make sure the test content targets were covered equally. Three experts from Universiti Sains Malaysia were selected to review the questionnaire to make sure that they were in the correct shape of the test. These experts had nothing to do with the study sample. All opinions and comments received from these experts were taken into account and used to further enhance the meaning, language and content of the questions.

A pilot study was conducted on 30 secondary school science teachers in Northern Malaysia where the teachers' abilities and school characteristics were similar to those of the teachers studied. Samples were selected from different schools to avoid pilot study contamination. This pilot study was conducted before the actual study in order to look at the suitability of the test in terms of content, timeliness of response and clarity of direction. Internal consistency tests were used to determine test reliability and the reliability coefficient used was Cronbach's Alpha. The index value was between 0 to1. The value of 0 denotes a low reliability level while the value of 1 denotes a high reliability level [25]. The alpha coefficient values obtained were 0.89. Therefore, the data from the study have high reliability. Hence, they were appropriate to be used as variables for the study. Five teachers were randomly selected and interviewed by the researcher using an interview framework by Braun and Clarke [26]. The data obtained were analyzed statistically using a descriptive SPSS software version 24.0.

## 3. Result

### 3.1. A. Demographic Information

Mean teacher age was 36.88 with a minimum of 26 years and a maximum of 50 years. In total, 21 male and 29 female teachers were involved in this study. There were 12 Chinese, 29 Malays, 8 Indians, and 1 Kadazan. Details on the information are summarized in Table 1.

**Table 1.** Demographic information.

| Features | | Frequency | % |
|---|---|---|---|
| Gender | Male | 21 | 42 |
| | Female | 29 | 58 |
| Race | Chinese | 12 | 24 |
| | Malay | 29 | 58 |
| | Indian | 8 | 16 |
| | Kadazan | 1 | 2 |
| Highest academic achievement | Bachelor degree | 40 | 80 |
| | Master | 10 | 20 |
| Option | Biology | 5 | 10 |
| | Physics | 9 | 18 |
| | Chemistry | 13 | 26 |
| | Mathematic | 2 | 4 |
| | Science | 19 | 38 |
| | Computer Science | 1 | 2 |
| | ICT | 1 | 2 |
| School grade | A | 29 | 58 |
| | B | 21 | 42 |
| School location | Urban | 26 | 52 |
| | Rural | 24 | 48 |
| Type of school | National secondary type school | 1 | 2 |
| | Religious secondary school | 3 | 6 |
| | National secondary school | 46 | 92 |
| State | Kedah | 20 | 40 |
| | Perak | 13 | 26 |
| | Perlis | 7 | 14 |
| | Penang | 10 | 20 |
| Teaching experience | >1–5 years | 6 | 12 |
| | >10 years | 34 | 68 |
| | >5–10 years | 10 | 20 |
| | >1–5 years | 6 | 12 |

### 3.2. B: FrogVLE Application

i. What is the frequency of using frogVLE among science teachers in North Malaysia while teaching secondary school science subjects?

Based on Table 2, the majority of teachers used frogVLE at least once a month (46%) and only 18% of teachers had never used frogVLE when teaching science. The others used frogVLE at least once a week (22%) and at least once a year (14%). The most frequent use of frogVLE for the experience of >10 years (16 teachers) and >1–5 years (5 teachers) was at least once a month. Teachers who taught between 5 and 10 years preferred frogVLE at least once a week (5 teachers). The frequency of the use of frogVLE among male teachers high, at almost 86%, while female teachers only 80%. The data show that almost 48% of teachers in grade A schools used frogVLE at least once a month, 24% at least once a week and 10% at least once a year. Meanwhile, 43% of grade B school teachers used frogVLE

at least once a month, 19% once aweek and 19% once a year. Based on the location, 50% of rural teachers used frogVLE once a month, 13% once a week and 13% once a year. In the urban area, 42% of teachers used frogVLE once a month, 31% once a week and 15% once a year. During the interview, respondents answered:

*"I use frogVLE as appropriate. Sometimes once a week. Sometimes once a month. Sometimes, I use frogVLE every time of Science. For example when I give the assignment to the student"* (R1). *"I use once a month. I also ask students to discuss in a forum outside of school time"* (R2). *"I use frogVLE once a week. But there are also my students playing frogplay at their home"* (R3). *"I only bring a student to the laboratory for frogVLEaccess when looking for information on the Internet, at least once a month"* (R4). *"I use frogVLE once a month. But students can send assignments via drive every time the task is completed"* (R5).

**Table 2.** Frequency of using frogVLEwhile teaching science.

| Features | | At Least Once a Month | At Least Once a Week | At Least Once a Year | Never |
|---|---|---|---|---|---|
| Overall | | 23 (46%) | 11 (22%) | 7 (14%) | 9 (18%) |
| Gender | Male | 12 (57%) | 5 (24%) | 1 (5%) | 3 (14%) |
| | Female | 11 (38%) | 6 (21%) | 6 (21%) | 6 (21%) |
| Highest academic achievement | Bachelor | 20 (50%) | 6 (15%) | 6 (15%) | 8 (20%) |
| | Master | 3 (30%) | 5 (50%) | 1 (10%) | 1 (10%) |
| School grade | A | 14 (48%) | 7 (24%) | 5 (17%) | 3 (10%) |
| | B | 9 (43%) | 4 (19%) | 4 (19%) | 4 (19%) |
| School location | Urban | 11 (42%) | 8 (31%) | 4 (15%) | 3 (12%) |
| | Rural | 12 (50%) | 3 (13%) | 3 (13%) | 6 (25%) |
| Teaching experience | >1–5 years | 5 (83%) | 0 (0%) | 0 (0%) | 1 (27%) |
| | >10 years | 16 (47%) | 6 (18%) | 6 (18%) | 6 (18%) |
| | >5–10 years | 2 (20%) | 5 (50%) | 1 (10%) | 2 (20%) |

ii. How much time is allocated each time to using frogVLE among science teachers in North Malaysia while teaching secondary school science subjects?

Based on Table 3, the highest frogVLE time duration was>30–60 min (36%) and the lowest was>5–15 min (14%). The highest frogVLE usage period based on experience was >30–60 min (>1–5 years: 3 people, >10 years: 11 people, >5–10 years: 4 people). The highest frogVLE use time frame based on gender was >30–60 min (male: 8 people, female: 10 people). The maximum duration of a master's degree was >5–15 min (4 people) while a bachelor's degree was >30–60 min (15 people). The highest frogVLE use time for grade A school was >30–60 min (14 people), while grade B was >15–30 min (8 people). For frogVLE teacher's use time in urban teachers, the highest was>30–60 min (12 people) and the lowest was0–5 min (3 people). For rural areas, the highest period was >15–30 min (8 people) and 0–5 min (8 people) while the lowest was >5–15 min (2 people). From the interviews, five teachers chose to allocate>30–60 min each time when using frogVLE. *"The use of frogVLE involves the use of the Internet and students need to access frogVLE in computer labs because there are no computers in the class. The relatively slow Internet access makes it difficult to access the app in frogVLE. The time of 30–60 min is quite good for learning using frogVLE. Even sometimes the time period is still inadequate"* (R3).

**Table 3.** The time allocated eachtime when using frogVLE.

| Features | | >15–30 | >30–60 | >5–15 | 0–5 |
|---|---|---|---|---|---|
| Overall | | 14 (28%) | 18 (36%) | 7 (14%) | 11 (22%) |
| Gender | Male | 7 (33%) | 8 (38%) | 2 (10%) | 4 (19%) |
| | Female | 7 (24%) | 10 (35%) | 5 (17%) | 7 (24%) |
| Highest academic achievement | Bachelor | 13 (33%) | 15 (38%) | 3 (8%) | 9 (23%) |
| | Master | 1 (10%) | 3 (30%) | 4 (40%) | 2 (20%) |
| School grade | A | 6 (21%) | 14 (48%) | 3 (10%) | 6 (21%) |
| | B | 8 (33%) | 4 (17%) | 4 (17%) | 5 (21%) |
| School location | Urban | 6 (23%) | 12 (46%) | 5 (19%) | 3 (12%) |
| | Rural | 8 (33%) | 6 (25%) | 2 (8%) | 8 (33%) |
| Teaching experience | >1–5 years | 2 (33%) | 3 (50%) | 0 (0%) | 1 (17%) |
| | >10 years | 8 (24%) | 11 (32%) | 7 (21%) | 8 (24%) |
| | >5–10 years | 4 (40%) | 4 (40%) | 0 (0%) | 2 (20%) |

iii. What are the frequently used applications in frogVLE among science teachers in North Malaysia while teaching secondary school science subjects?

The results in Table 4 show that the most used applications were quizzes (36%) while the least used were tutorials (2%) and sites (2%). The other applications are email (10%), forum (8%), discover (6%), video (6%), frogplay (6%), assignment (4%), bookshelf (4%), chat (4%), and drive (4%). The interview data clearly shows that only a few applications were used often: "*I use quizzes and assignment as it is easier to use. Other applications are rarely used or almost indirect because of lack of exposure and weakness in ICT*" (R1). "*I prefer to use the forum because it does not need a neat setup*" (R2). "*I used a lot of frogplay because my students prefer learning while playing*" (R3). "*I showed a learning video using the discover app. Students also search for information through discovering applications*" (R4). "*I prefer the drive to save the assignment provided by the student. It saves cost and is easily accessible for review*" (R5).

**Table 4.** Frequently used applications.

| Application | Frequency | % |
|---|---|---|
| Assignment | 2 | 4 |
| Bookshelf | 2 | 4 |
| Chat | 2 | 4 |
| Discover | 3 | 6 |
| Drive | 2 | 4 |
| Email | 5 | 10 |
| Forum | 4 | 8 |
| Frogplay | 3 | 6 |
| Quiz | 18 | 36 |
| Sites | 1 | 2 |
| Null | 4 | 8 |
| Tutorial | 1 | 2 |
| Video | 3 | 6 |

iv. What are the issues and challenges of frogVLE among science teachers in North Malaysia while teaching secondary school science subjects?

The main issues and challenges of applying frogVLE in science teaching (refer to Table 5) are an inadequate number of computers (30%), the problem of access to frogVLE applications was relatively slow (22%), Internet access problems (20%) and time constraints (18%). The other challenges were that students only accessedfrogVLE in schools (2%) or didnot even access it at all (2%). The interview data show that there are still many issues related to frogVLE in schools.

*"My school computer lab is only one but my schoolboys are crowded. It is a struggle to bring students to a computer lab to access frogVLE. It is more unfortunate that Internet access is somewhat slow and it makes it difficult for students to access applications in frogVLE"* (R1). *"Teachers at my school are aware of the importance of using frogVLE. But due to a lack of knowledge and ICT skills teachers cannot use frogVLE applications while teaching science"* (R2). *"Clerical work at school is huge. The use of frogVLE requires initial planning. With limited time, most teachers prefer to spend on the syllabus for the exam using easier teaching materials besides frogVLE"* (R3). *"I always use frogVLE while teaching. But, frogVLE is less user-friendly. Its time to load every app in frogVLEis very slow. To use apps also requires ICT skills and is quite challenging for a rather weak student"* (R4). *"The issue in my school is more to the lack of ICT facilities and less ICT skills"* (R5).

**Table 5.** Issues and challenges faced when using frogVLE.

|  | **Frequency** | **%** |
|---|---|---|
| Access to applications in frogVLE is quite slow | 11 | 22 |
| Not yet mastered because there are no viewers/users to see the effectiveness of the frogVLE medium in science teaching | 1 | 2 |
| Not all students can access the Internet at home | 3 | 6 |
| Internet access and computer facilities in science labs are not satisfactory | 1 | 2 |
| Low Internet access | 5 | 10 |
| Unstable Internet connection | 1 | 2 |
| Smart teacher does not show a role. Teacher Activity Centre also does not take responsibility. The regional committee also does not use frogVLE applications. | 1 | 2 |
| The number of computers is not enough | 11 | 22 |
| Facilities constraint for students in schools because only one computer laboratory is in school for all. At home, students do not have Internet access. | 4 | 8 |
| Time constraint | 9 | 18 |
| Students access frogVLE in schools only | 1 | 2 |
| Students do not access frogVLE | 1 | 2 |
| Not suitable for use | 1 | 2 |

v. What are the suggestions for improvements in frogVLE among science teachers in North Malaysia while teaching secondary school science subjects?

Table 6 shows that 58% of respondents suggested that there is a need to increase information, technology and communication (ICT) facilities inschools in order to improve the frogVLE application in science teaching. In total, 34% of respondents wanted an improvement in the frogVLE application itself to make it more user friendly. In total, 2% of respondents wanted training and courses on frogVLE. A total of 2% of respondents said that research on the effectiveness of frogVLE should be done. Suggestions by the interviewed teachers are as follows:

*"The government needs to upgrade ICT facilities at schools and provide a complete set of computers with enough Internet for all students"* (R1). *"Higher quality continuous training is required according to the level of knowledge and skill of the teacher"* (R2). *"Clerical work should be reduced"* (R3). *"The relevant parties need to improve frogVLE to be more user friendly. Students are also given courses to improve ICT skills"* (R4). *"ICT facilities in schools need to be improved and a key instructor should be placed in each school to assist if there is a problem with frogVLE"* (R5).

**Table 6.** Suggestions for improvements in frogVLE while teaching secondary school science subjects.

|  | Frequency | % |
|---|---|---|
| Provide more computers at school | 12 | 24 |
| Supply laptop with Internet to all students | 4 | 8 |
| Provide tablets with Internet packages to school labs to be used by all students | 1 | 2 |
| Computer facilities are extended to classrooms and science labs | 2 | 4 |
| FrogVLE should be more user friendly | 3 | 6 |
| More exposure is given to teachers onfrogVLE | 1 | 2 |
| Choose a pilot group (1 teacher: 30 students) to see the effectiveness of frogVLE and become a mentor for another student's teacher | 1 | 2 |
| Make access to every application easier | 6 | 12 |
| Make it easy to generate sites, quizzes and more so IT teachers do not easily get bored | 1 | 2 |
| Diversifying teaching aids in frogVLE | 2 | 4 |
| Provide more computer labs /access rooms | 1 | 2 |
| Use smart phone to access frogVLE | 1 | 2 |
| Provide ready made questions and notes on frogVLE | 2 | 4 |
| Button icon in frogVLE should be more user friendly. As for now, always need to import and capture pictures of symbols for chemistry. If need to import and link then for me frogVLEis rather burdensome. | 1 | 2 |
| Increase Internet speed | 8 | 16 |
| No suggestion | 3 | 6 |
| FrogVLE should not be used during teaching | 1 | 2 |

## 4. Discussion

The result from the study shows how frogVLE applications were used in teaching science. Secondary school science teachers in North Malaysia used frogVLE but could not use them optimally due to some unresolved issues. The frequency of using frogVLE while teaching science and the time allocated each time when using frogVLE was low when compared to the number of times for teaching science in a year [2]. This finding is supported by the previous studies conducted to determine the use of ICT in teaching [8,9,12,13,19,20,27,28]. Teachers only used some of the applications found in frogVLE when teaching science while there are more applications that can be explored and applied to create interactive and engaging learning. This may be due to limited time and lack of knowledge and ICT skills. Most teachers have a modest level of knowledge and skills in which they are familiar with application software such as word processing and electronic presentations but not Internet and email applications [11,13].

This study also shows that there are many issues and challenges of frogVLE while teaching secondary school science subjects in Malaysia.The main challenges of applying frogVLE in science teaching is the lack of ICT facilities where there is an inadequate number of computers and Internet access problems, access to frogVLE applications is relatively slow, and time constraint, which are issues that are interconnected. The other challenge is that students only access frogVLE in schools or do not even access it at all. The findings are consistent with previous studies [9,12,15,19,20,27,28]. A survey was done by Termit Kaur and Samli [27] to investigate the knowledge level, attitude towards the use of ICT in teaching and learning, and obstacles faced by 50 in-service teachers in secondary schools in the state of Penang. The results show that teachers had a strong desire to integrate ICT in education. However, they faced many obstacles. Therefore, immediate action should be taken because the teachers' well-equipped preparation with ICT tools and facilities is one of the main factors in the success of technology-based teaching and learning especially in frogVLE [8,9]. According to Garba et. al. [12], there are two things necessary in building the most desirable 21st century learning environment for our digital age of information (21st century students). First, provide the basic infrastructure and ICT facilities and get teachers to use these facilities in teaching and learning. The second step is getting teachers to adopt the use of the 21st century.

Unfortunately, previous studies have shown that there is an increase in the use of ICT by teachers but not in teaching. Teachers actively access the Internet for personal use such as email, social networking, student records and to access information that helps to prepare lesson plans and preparations to discover a limited knowledge of TPACK, but rapid administrative and technology issues emerge as challenges that prevent teachers from using the facilities in their teaching in aMalaysian context [12]. There are also teachers who have knowledge but do not use them because of time constraints. Early planning needs to be done to integrate in teaching and at the same time, teachers need to complete all the topics into the syllabus in preparation for exams and many outdoor activities [20,27,29]. This situation is explained by the study conducted by Awang et. al. [30] showing that there is a relationship between workload and the desire to use frogVLE. The findings indicate that while teachers are fairly proficient in their computer and internet skills and have fairly high computer self-efficacy, their workload and a structured and standardized curriculum were inhibitors of Web 2.0 adoption [31]. There are even teachers who do not face time constraints and lack of skills and knowledge but still have a negative attitude towards the use of frogVLE in teaching. Some teachers feel less confident about their skills indigital technology so that it affects their attitude toward the use of digital technology in their teaching [14].

Findings about inadequate computer hardware and access give the message that the government's efforts to equip ICT facilities in schools are still inadequate and require improvement [9]. The results of this study show that the target to provide high-speed internet access and a virtual learning platform (frogVLE) to bridge the digital divide between urban and rural areas had not yet been achieved [2]. According to Mei et al. [9], an average school still only has one computer lab with limited computers and Internet access, leaving only one class to access frogVLE at one time. Teachers are forced to scramble to use computer laboratories for teaching, causing them to be less interested in applying frogVLE [9]. Previous studies have found that most of the selected schools have at least basic ICT resources in the science classrooms, but they are not used optimally. It was also concluded that the perceived usefulness of ICT resources seemed to be the most influential factor for the teachers' intention to practice computer-assisted learning [11,13,19,27]. This problem may be overcome if every student is supplied with a computer equipped with high-speed Internet packages to enable them to access frogVLE anywhere at anytime.

There is also a need to improve the frog VLE application itself to make it more user friendly. When users of a new VLE technology struggle with a particular functionality, easily accessible, clear and searchable support materials are important factors in accepting a VLE [5]. Research on the effectiveness of frogVLE and how to integrate it into teaching should be done.The findings must be shared to teachers to improve their teaching using frogVLE. To date, teachers in Malaysia are still need intensive training in the use of Information Technology (IT) to facilitate their integration into classroom activities [27]. They claimed the workload in school prevented them from exploring and mastering the system further. As such, most of the teachers expressed acritical need for more training and exposure to the system [9]. It was also found that the attitudes of teachers regarding the use of ICT vary with their years of experience and their level of knowledge on ICT [27]. Therefore, ICT and frogVLE courses need to be improved in order to enhance their skills and knowledge using frogVLE in teaching. Teachers also have to improve their teachings and teaching environment to meet the needs of their pupils and the curriculum for 21st century learning [13].

## 5. Conclusion and Implication

The findings of this study are expected to help the Ministry of Education to develop programs and improve the use of a virtual learning environment (VLE) such as the frogVLE application in teaching science in secondary schools. Monitoring in schools needs to be improved to determine the exact level of VLE implementation to ensure that VLE is used in accordance with the established guidelines and goes smoothly. The issues raised should be addressed urgently to avoid aworse situation. For future research, studieson the effectiveness of frogVLE in science and how to integrate it in teaching should

be done. Structured interviews and direct observations in the classroom, examination of teaching record books and teaching aids of use, as well as records of ICT usage and computer labs need to be done and data collected to find out the real situation in school. The views of all parties including administrators, students and parents should also be taken into account. As VLE has a lot of benefits to science education, the research should be broadened to other countries in order to get a real picture on how VLE is implemented in schools around the world and action should be taken to improve it. In conclusion, a virtual learning environment such as the frogVLE application can only be used optimally to help improve science teaching excellence if therelevant issues are solved and supportis receivedfrom all parties.

**Author Contributions:** Conceptualization, N.R. and S.S.; Formal analysis, N.R.; Investigation, N.R.; Methodology, N.R. and S.S.; Supervision, S.S.; Writing – original draft, N.R.; Writing – review & editing, N.R. and S.S.

**Funding:** This research received no external funding.

**Conflicts of Interest:** The authors declare no conflicts of interest.

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
