# Peer review of "FrogVLE Application in Science Teaching in Secondary Schools in North Malaysia: Teachers’ Perspective"

_education, doi:10.3390/educsci9040262_

Round 1
Reviewer 1 Report
This is indeed very interesting and well presented work. However, there are a few issues which need to be addressed and here they are.
You mention that the questionnaire included 20 items but we don't know what they are about.Â
In the paper you inform us of part A - demographic part - but there is no mentioning of part B.Â
You mention the findings but you discuss the answers of the research questions. It would be clearer for the reader to have a results section that presents the answers to the questions and the percentage of their final output and then discuss the findings per research question in a discussion section.
Finally, it is not clear whether there was a piloting of the questionnaire....
The strong points of this paper are the excellent organization, the effort to follow the scientific research protocols and the good linguistic aspect.Â
It is certain, that if the above issues are addressed, the paper will be of much interest to the readers
Best of luck!
Author Response
Dear reviewer,
Thank you for your comments. I try my best to do corrections based on your suggestions.
The details are as attached.
Kind regards,
Nofaizah Ramli
Â

Reviewer 2 Report
I think the study is well done, however, the findings are probably limited to those interested in higher education in Malaysia. I think this submission will probably not be of interest to readers in general.
Specific comments are as follows:
1. There is little description of what frogVLE can actually do until the Discussion, when some of its features are mentioned in a very brief list. It would be helpful to readers outside Malaysia to know what sort of features frogVLE provides for teachers.
2. ICT is used without any indication of what it stands for. Please define it at least once before referring to it as "ICT".
3. The writing seems to be for a Malaysian audience, as there are numerous instances where frogVLE and the word "teachers" are used, without reference specifically to Malaysia. If this were a submission to a Malaysian journal, that would be fine, but for an international journal, making statements about teachers comes across as meaning teachers, everywhere, in general, as opposed to teachers in Malaysia specifically.
4. The English is okay. There are a few instances where, as a native English speaker, I couldn't quite figure out exactly what the authors meant, and as a reader, I do not want to misinterpret what the authors are trying to convey. Examples: line 88, line 260.
5. Regions of Malaysia are mentioned a number of times, without any reason as to why analyzing the data according to region is important. I guess that perhaps there are demographic differences between the regions, but these are not mentioned at all.
6. Race is mentioned as a characteristic the authors looked at in their data, but there is no indication as to why. For western readers, that will probably seem strange. If the authors had a good reason for this, or in Malaysia this is important for their data analysis, the authors should state that fact so there is not confusion.
7. The Conclusion summarizes the findings, but also seems to suggest that this study is just a preliminary one, and that a more intensive study needs to be carried out later. If that's the case, the questions must be asked: "Did the authors find anything significant enough in this study to share with an international audience?" Perhaps if the authors wrote for a more general audience, the answer would be yes, but as it is, this piece is probably only of interest to Malaysian educators. Could the authors broaden the article so it is of more interest to an international audience? Of course, if other reviewers disagree with this view, that's fine--perhaps the article just seems too narrowly focused to me, and others would find it of more interest.
8. Data (numerical findings) should perhaps be summarized in tables, as well as discussed in the text. Numbers can get lost in the text, and it was sometimes difficult to follow the numerical findings.
Author Response
Dear reviewer,
Thank you for your comments. I try my best to do corrections based on your suggestions.
2. The details are as attached.
Â
Kind regards,
Nofaizah Ramli

Reviewer 3 Report
This manuscript addresses the implementation in the real classroom of the frogVLE, a virtual learning environment launched by the Malaysian Ministry of education in an attempt to innovate education in the country.
The paper includes a descriptive analysis of the usage, difficulties, and affordances of the platform, and it's intended to inform eventual improvements to be undertaken by the Ministry of education. This is an interesting approach, but the manuscript has some problems that should be solved before it can be considered for publication.
The results include many different comparisons and are difficult to read. In these conditions is difficult to make sense of the data. In my view, the authors should consider: Using a table and/or graphics to summarize the information. Focus on the more relevant comparisons, those where significant differences are found, or those about which a sensible hypothesis can be formulated. The methodology and more specifically the structure and content of the questionnaire should be openly published. Note that the Crombach's alpha is an indicator of reliability, but not of validity. validity and reliability are two different psychometric properties, and must be assessed by different means. Spread over the text there are references to the digital competence of teachers, their use of ICT, the quality and quantity of the infrastructures, etc. The most recent of these references are 5 years old. ICT in education are a rapidly evolving field, and I would day these indicators are quite likely to have evolved since these papers where published. I would encourage the authors to included more updated references, also from other non-local contexts. The introduction should be more explicit about the nature, possibilities and intended use of the frogVLE. Without knowing the platform it's difficult to make certain inferences. The English should be revised and edited all over the paper. Likewise, there are many spelling errors or typos, which should be corrected. The discussion returns once and again to the same ideas. At times, it's difficult to undestand whetehr the author(s) are making refence to this study or to a different one. Please note that there is growing evdence that providing infrastuructures is not enough for granting pedagical integration or impactful use of the ICT. Define what would be an optimal use of the platformÂ
Author Response
Dear reviewer,
Thank you for your comments. I try my best to do corrections based on your suggestions.
2. Details are as attached.
Kind regards,
Nofaizah Ramli

Round 2
Reviewer 3 Report
Dear Author,
I'm glad to see most of my suggestions have been incorporated into the manuscript. Although the results section is still a bit cumbersome, it is now much more readable, thanks to the inclusion of tables and edition of the English.
Some new recent references have been included, that maybe not reflect the wealth of related research published in the last years, but at least convey more up-to-date information.
I reiterate my previous suggestion to "focus on the more relevant comparisons, those where significant differences are found, or those about which a sensible hypothesis can be formulated". With so many factors involved, it's hard to get the whole picture. Specially worrisome is the use of absolute numbers/ relative frequencies in tables. The usage should be consistent (1-3 only freq, 4 only %, 5 both freq and %), and I would definitively opt for the option freq + %. Otherwise, using only freqs may be misleading, specially when groups are imbalanced (as, for example, in the case of Master vs. Bachelor; 10 vs 40 individuals).
Please, double checkcarefully your manuscript for typos (gaps between words missing, repeated sentences in lines 130-150...).
Author Response
Dear reviewer,
Thank you for your comment. I have tried my best to do corrections based on your suggestions. Details are as attached.
(Please see the attachment)
